# Adjuvant Transarterial Chemoembolization Following Curative-Intent Hepatectomy Versus Hepatectomy Alone for Hepatocellular Carcinoma: A Systematic Review and Meta-Analysis of Randomized Controlled Trials

**DOI:** 10.3390/cancers13122984

**Published:** 2021-06-15

**Authors:** Stepan M. Esagian, Christos D. Kakos, Emmanouil Giorgakis, Lyle Burdine, J. Camilo Barreto, Michail N. Mavros

**Affiliations:** 1Oncology Working Group, Society of Junior Doctors, 15123 Athens, Greece; stepesag@sni.gr; 2Surgery Working Group, Society of Junior Doctors, 15123 Athens, Greece; chriskakos@auth.gr; 3Department of Surgery, University of Arkansas for Medical Sciences, Little Rock, AR 72205, USA; egiorgakis@uams.edu (E.G.); lburdine@uams.edu (L.B.); jcbarretoandrade@uams.edu (J.C.B.)

**Keywords:** transarterial chemoembolization, lipiodol, hepatic artery, embolization, liver resection, hepatic resection

## Abstract

**Simple Summary:**

The role of postoperative transarterial chemoembolization (TACE) after liver resection for hepatocellular carcinoma (HCC) remains unclear. We performed a systematic review of the literature and compared the survival outcomes of TACE vs. no TACE after liver resection for HCC. While the existing evidence suggests a benefit of adjuvant TACE, published trials to date are at significant risk of bias and limited to Eastern Asian patients. High-quality randomized clinical trials are needed to confirm the oncologic benefits of adjuvant TACE.

**Abstract:**

The role of adjuvant transarterial chemoembolization (TACE) for patients with resectable hepatocellular carcinoma (HCC) undergoing hepatectomy is currently unclear. We performed a systematic review of the literature using the MEDLINE, Embase, and Cochrane Library databases. Random-effects meta-analysis was carried out to compare the overall survival (OS) and recurrence-free survival (RFS) of patients with resectable HCC undergoing hepatectomy followed by adjuvant TACE vs. hepatectomy alone in randomized controlled trials (RCTs). The risk of bias was assessed using the Risk of Bias 2.0 tool. Meta-regression analyses were performed to explore the effect of hepatitis B viral status, microvascular invasion, type of resection (anatomic vs. parenchymal-sparing), and tumor size on the outcomes. Ten eligible RCTs, reporting on 1216 patients in total, were identified. The combination of hepatectomy and adjuvant TACE was associated with superior OS (hazard ratio (HR): 0.66, 95% confidence interval (CI): 0.52 to 0.85; *p* < 0.001) and RFS (HR: 0.70, 95% CI: 0.56 to 0.88; *p* < 0.001) compared to hepatectomy alone. There were significant concerns regarding the risk of bias in most of the included studies. Overall, adjuvant TACE may be associated with an oncologic benefit in select HCC patients. However, the applicability of these findings may be limited to Eastern Asian populations, due to the geographically restricted sample. High-quality multinational RCTs, as well as predictive tools to optimize patient selection, are necessary before adjuvant TACE can be routinely implemented into standard practice. PROSPERO Registration ID: CRD42021245758.

## 1. Introduction

Among all cancers, liver cancer currently ranks sixth in terms of incidence and third in terms of cancer-related mortality worldwide. Hepatocellular carcinoma (HCC) represents the most common form of primary liver cancer (75–80%) [1]. The Barcelona Clinic Liver Cancer (BCLC) staging system is used to classify HCC into four broad categories (early (A), intermediate (B), advanced (C), and terminal (D) stages) [2]. Early-stage (BCLC A) HCC patients can benefit from curative treatments, including resection, transplantation, and ablation therapies [3]. Patients with a solitary tumor and well-preserved baseline liver function (Child-Pugh A; no portal hypertension) are considered to be optimal surgical candidates for hepatectomy with curative intent [4,5]. Patients who do not qualify for surgical resection and have a limited disease burden, as defined by various selection criteria (e.g., Milan criteria, extended Toronto criteria, University of California San Francisco criteria), may receive liver transplantation [6,7,8]. Patients with small tumors in whom surgery might entail significant morbidity may opt for ablation therapy instead [4], even though surgery is associated with superior oncological outcomes [9]. Despite the fact that many patients may not qualify as optimal surgical candidates due to impaired liver function or multinodular disease, resection is still considered the primary therapeutic modality for early-stage HCC in most cases, as the scarcity of liver donors has resulted in long waiting lists for transplantation [3].

Tumor recurrence following hepatectomy remains a major hurdle in the successful management of early-stage HCC. Even for small tumors, the 5-year recurrence rate following resection is almost 70% [10]. A variety of strategies employing adjuvant therapeutic modalities (both systemic and locoregional) have been proposed over the years, aiming to reduce recurrence following resection with varying success. One of them is transarterial chemoembolization (TACE), which is also currently the standard of care for intermediate-stage disease (BCLC B) [4,5,11]. Other options have included interferon [12], antiviral therapy [13], sorafenib [14], radioembolization with I^131^ [15], and systemic chemotherapy [16]. Radioembolization with Y^90^ is increasingly used for unresectable or borderline resectable tumors as well [17,18]. The concept of adjuvant TACE after hepatectomy for resectable HCC is not new; multiple studies (including numerous randomized controlled trials (RCTs)) spanning multiple decades have attempted to provide a definitive answer, but with little success and often conflicting results.

In this context, to clarify the role of adjuvant TACE in the setting of resectable HCC, we performed a systematic review of the available high-quality evidence (RCTs) and synthesized their data using the methodology of meta-analysis.

## 2. Materials and Methods

### 2.1. Study Design and Inclusion/Exclusion Criteria

This systematic review was performed according to the Preferred Reporting Items for Systematic Reviews and Meta-Analyses (PRISMA) guidelines and in line with the protocol agreed upon by all authors prior to the beginning of this study (Appendix A) [19]. The protocol of the study was submitted to the international prospective registry for systematic reviews PROSPERO (ID CRD42021245758) (Appendix A). An Institutional Review Board approval or patient written consent was not required, as we only used data from already published studies.

The inclusion criteria were defined by applying the Population/Participants, Intervention, Comparison, Outcome and Study design (PICOS) framework:Participants: Patients of any age, sex, or race with resectable HCC undergoing hepatectomy with curative intentInterventions: Adjuvant TACE following hepatectomyComparison: Hepatectomy aloneOutcomes: Overall survival (OS), recurrence-free survival (RFS)Study design: Randomized controlled trials (RCTs)

Between studies with overlapping populations, we only included the most recent one or the one reporting on the largest number of patients. Excluded studies met at least one of the following criteria: (i) papers published in any language other than English, Spanish, French, Italian, or Chinese; (ii) animal studies; (iii) non-randomized retrospective or prospective clinical studies; (iv) case reports and case series; (v) narrative or systematic reviews and meta-analyses; (vi) editorials, comments, opinions, letters to the editor, published abstracts, errata, and trial protocols; (vii) studies on unresectable HCC or palliative hepatectomy; and (viii) studies that included neoadjuvant therapeutic interventions (either systemic or locoregional) prior to hepatectomy.

### 2.2. Literature Search Strategy

Eligible studies were identified through a comprehensive search of the MEDLINE (through PubMed), Embase, and Cochrane Library bibliographic databases, as well as the clinical trial registry Clinicaltrials.gov (accessed on 14 January 2021), using the following search algorithm: (hepatocellular carcinoma OR HCC OR hepatoma OR liver cancer OR hepatic cancer or hepatocarcinoma) AND (transarterial chemoembolization or transcatheter arterial chemoembolization or TACE or lipiodol) AND (resectable OR adjuvant OR post-operative OR hepatectomy OR liver resection OR hepatic resection). Two independent reviewers (S.M.E., C.D.K.) initially screened the titles and abstracts and then the full texts of all relevant articles using the systematic review software Covidence [20]. Any discrepancies were identified and resolved through quality control discussions with the senior author (M.N.M.). We also manually searched the references of all included articles and previously published narrative and systematic reviews for any that were missed but otherwise eligible according to the systematic “snowball” methodology [21].

### 2.3. Data Tabulation and Extraction

A standardized, pre-piloted Microsoft Excel form was used to tabulate and extract data from the included studies for the synthesis of evidence. Two reviewers (S.M.E., C.D.K.) extracted all data independently and any disagreements were identified and resolved by reaching a consensus or through quality control discussions with the senior author (M.N.M.). We extracted the following data from all included studies: (i) study characteristics (first author, year of publication, study center, country, dates of enrollment, number of patients for each group); (ii) baseline variables (gender, age in years, BCLC stage, Eastern Cooperative Oncology Group performance status, Child–Pugh score (CPS) or Model for End-Stage Liver Disease (MELD) score if CPS was not available, HBV/HCV virology status, bilirubin level in mmol/L, prothrombin time in seconds, international normalized ratio, tumor size in cm, tumor number, alpha-fetoprotein (AFP) in ng/mL, microvascular invasion (MVI), portal vein tumor thrombosis (PVTT)); (iii) operative/postoperative variables (hepatectomy technique (anatomic vs. parenchymal-sparing), other locoregional treatment following tumor recurrence (TACE, hepatic artery infusion pump chemotherapy, transcatheter arterial embolization, radioembolization, radiofrequency ablation, microwave ablation, cryoablation), interval between HCC resection and TACE, TACE regimen (agent, dose and frequency), and adverse events attributable to TACE (using the definitions provided by authors and graded according to the US National Cancer Institute’s Common Terminology Criteria for Adverse Events (CTCAE) version 4.0)); and (iv) outcomes (OS, RFS).

### 2.4. Risk of Bias in Individual Studies

We assessed the risk of bias using Cochrane’s Risk of Bias 2.0 (RoB 2.0) tool for randomized trials. Using prespecified signaling questions for guidance, a risk of bias assessment is made for the following five domains: (i) randomization process, (ii) deviations from intended interventions, (iii) missing outcome data, (iv) measurement of the outcome, and (v) selection of the reported result. For each domain, the risk of bias can be characterized as “Low risk”, “Some concerns”, or “High risk”. Based on the risk of bias of each individual domain, an overall risk of bias judgment is made for each study [22].

### 2.5. Statistical Analysis

#### 2.5.1. Data Pooling

Categorical variables were presented as frequencies and percentages. Continuous variables were presented as means and standard deviations (SDs). When continuous data were provided as medians and ranges, we used the methods described by Hozo et al. to estimate the respective means and SDs [23]. When the included studies reported on the medians and interquartile ranges, we used the method described by Wan et al. to estimate the respective means and SDs [24]. All relative rates were estimated based on the available data for each variable of interest and all available data were handled according to the principles stated in the Cochrane Handbook [25].

#### 2.5.2. Meta-Analysis

Meta-analysis was carried out to compare the outcomes of adjuvant TACE vs. no adjuvant treatment after hepatectomy for resectable HCC. Time-to-event outcomes were summarizing using hazard ratios (HRs) and their 95% confidence intervals (CIs); a HR > 1 corresponded to a greater risk of the event in the adjuvant TACE group. When HRs and their 95% CIs for time-to-event outcomes were not provided, we used the methods described by Tierney et al. to estimate them from the respective Kaplan–Meier survival curves [26]. Between-study heterogeneity was assessed using the Cochran Q statistic and by estimating *I^2^*. High statistical heterogeneity was confirmed with a significance level of *p* < 0.05 and *I^2^* ≥ 50%. The pooled estimates were calculated using the random-effects model (DerSimonian–Laird) to take into account potential between-study clinical heterogeneity and to adopt a more conservative approach [27]. The overall quality of evidence was assessed using the GRADE approach. Publication bias was evaluated using funnel plots and Egger’s formal statistical test [28,29]. For the interpretation of Egger’s test, statistical significance was defined as *p* < 0.1. Statistical significance was set at *p* < 0.05 for all other comparisons and all *p*-values were two-tailed. Statistical analyses were performed using Stata IC 16.0 (StataCorp LLC, College Station, Texas, TX, USA).

#### 2.5.3. Additional Analyses

We planned a priori random-effects meta-regression analyses and subgroup analyses for the following variables as potential confounders, effect modifiers, and sources of heterogeneity: (i) HBV/HCV virology, (ii) MVI, (iii) PVTT, (iv) tumor size, (v) tumors fitting Milan criteria, (vi) cirrhosis (CPS and/or MELD score), (vii) hepatectomy technique (anatomical vs. parenchymal-sparing), (viii) trial location (Eastern vs. Western countries), and (ix) risk of bias (low/some concerns vs. high). For categorical variables, meta-regression was conducted according to the percentage of the variable in the total population of each study. For continuous variables, meta-regression was conducted according to the standardized mean difference of the variable among the two groups. We only performed meta-regression if at least five studies reported data on the variable of interest, as the statistical power of the analysis would otherwise be compromised.

## 3. Results

### 3.1. Study Selection and Characteristics

The initial literature search yielded 3911 potentially relevant records. After screening titles and abstracts, 87 articles were retrieved for full-text evaluation. Overall, 10 studies [30,31,32,33,34,35,36,37,38,39] reporting on 597 patients with resectable HCC who underwent hepatectomy plus adjuvant TACE and 619 who underwent hepatectomy alone satisfied our inclusion criteria and were included in this systematic review (Figure 1). An additional two RCTs that fulfilled our inclusion criteria were also identified but were ultimately excluded as their data did not permit proper hazard ratio extraction for any of the outcomes [40,41]. Nine studies were from China [31,32,33,34,35,36,37,38,39] and one was from Japan [30]. The included studies were published between 1994 and 2018 and included patients were enrolled between January 1987 and August 2014. All ten studies were conducted in a single-center setting [30,31,32,33,34,35,36,37,38,39]. The study characteristics of all included RCTs, along with the patient baseline characteristics, are presented in Table 1.

### 3.2. Risk of Bias Assessment

We assessed the individual risk of bias of the 10 randomized controlled trials using the RoB 2.0 tool. The overall risk of bias was determined to be high for four studies [33,34,35,37], whereas there were some concerns for the remaining six studies [30,31,32,36,38,39]. In particular, Li et al. and Li et al. were considered to be at high risk of bias due to missing outcome data (attrition bias), as a subset of eligible patients were excluded from the analysis due to postoperative complications related to the hepatectomy procedure [33,34]. Peng et al. was characterized to be at high risk of bias due to missing outcome data because a significant proportion of patients (>20%) were lost to follow-up and missingness could be related to the outcome (e.g., treatment toxicity, disease progression, or death) [35]. Wei et al. was considered to be at high risk of bias due to deviations from the intended interventions, as two patients randomized to the TACE group withdrew their consent and therefore did not receive TACE within the trial (i.e., fear of toxicity) [37]. There were some concerns for the remaining studies, mainly stemming from potential bias in the randomization process, deviations from the intended interventions, and selection of the reported result (Figure 2).

### 3.3. Synthesis of Results

The results of our meta-analyses for both outcomes and the quality of evidence presented in this study were summarized using the GRADE approach (Appendix A).

#### 3.3.1. Overall Survival

Seven studies reported on patient OS [30,31,32,35,36,37,39]. Hepatectomy plus adjuvant TACE was associated with a significantly lower risk of death compared to hepatectomy alone (HR = 0.66, 95% CI, 0.52 to 0.85; *p* < 0.001; *I^2^* = 37.3%; Figure 3). There was no evidence of publication bias upon visual inspection of the funnel plots and based on Egger’s test (*p* = 0.38) (Appendix A).

#### 3.3.2. Recurrence-Free Survival

Nine studies reported on RFS [30,31,32,33,34,36,37,38,39]. Hepatectomy plus adjuvant TACE was associated with a significantly lower risk of recurrence compared to hepatectomy alone (HR = 0.70, 95% CI: 0.56 to 0.88; *p* < 0.001; *I^2^* = 52.3%; Figure 4). There was no evidence of publication bias upon visual inspection of the funnel plots and based on Egger’s test (*p* = 0.91) (Appendix A).

#### 3.3.3. Recurrence Patterns

Four studies reported on the recurrence location patterns of each group [30,31,36,39]. Based on the available data, both groups had similar recurrence patterns, with the majority being intrahepatic. Pooling the four studies, in the adjuvant TACE group, the intrahepatic recurrence rate was 59.4% (111/187), the extrahepatic recurrence rate was 16.0% (30/187), and 24.6% (46/187) of patients had both intrahepatic and extrahepatic disease recurrence. Similarly, the intrahepatic recurrence rate in the hepatectomy only group was 65.3% (115/176), the extrahepatic recurrence rate was 12.5% (22/176), and 22.2% (39/176) of patients had both intrahepatic and extrahepatic disease recurrence.

#### 3.3.4. Adverse Events

Eight out of ten studies reported at least some information on the adverse effects attributable to TACE [30,32,33,34,35,36,37,42]. However, there was significant heterogeneity in the reporting methods and definitions used by the authors of each study, precluding us from pooling their results in most cases. Based on the available data, 42.1% (*n* = 83/197) of patients experienced a Grade I/II [36,42] and 2.7% (*n* = 6/220) a Grade III/IV/V [30,42] increase in hepatic enzymes (aspartate and alanine aminotransferase). Hypoalbuminemia occurred in a total of 4.2% (*n* = 13/313) of patients, with all of these cases being Grade I/II [36,37,42]. The hyperbilirubinemia rate was 22.0% (*n* = 69/313) [36,37,42], with all cases being characterized as Grade II or lower. Aside from one case of Grade III liver bleeding, reported by Wei et al. [37], no other Grade III/IV/V complication was reported by any of the authors.

#### 3.3.5. Additional Analyses

We performed meta-regression for the following categorical variables according to their prevalence in the total population of each study: HBV, MVI, and anatomic hepatectomy for both OS and RFS (Appendix A). We also performed meta-regression for RFS according the to the mean tumor size in each study (Appendix A). None of these variables had a significant influence on either OS or RFS. We also performed a subgroup analysis according to the risk of bias among the included studies; TACE was associated with superior OS and RFS in both studies at high risk of bias, as well as studies with some concerns for bias (Appendix A). We were unable to perform meta-regression according to mean tumor size for OS, or for HCV status, PVTT, tumors fitting the Milan criteria, or cirrhosis status for either of the outcomes, as less than five studies reported data on each of these variables. In addition, the pre-planned subgroup analysis according to study location was not feasible, as all studies were conducted in Eastern Asia.

## 4. Discussion

The results of this systematic review and meta-analysis suggest that adjuvant TACE may have a role in the treatment of resectable HCC. The combination of adjuvant TACE and hepatectomy was associated with superior OS and RFS compared to hepatectomy alone. However, no definitive conclusions or recommendations can yet be made, as the current available evidence remains weak and at significant risk of bias.

Numerous meta-analyses in the past have attempted to clarify the role of adjuvant TACE for resectable HCC [42,43,44,45,46,47,48,49,50,51,52,53]. Among these, only Zhong et al. and Cheng et al. limited their inclusion criteria to RCTs [42,44]. RCTs represent a higher level of evidence compared to non-randomized studies because they are less likely to yield biased effect estimates. For this reason, current systematic review and meta-analysis guidelines recommend that the inclusion criteria in meta-analyses comparing interventions should be limited to RCTs alone, whenever feasible [54,55]. Cheng et al. synthesized data from RCTs investigating the role of TACE in both the neoadjuvant and adjuvant setting and so a direct comparison with our meta-analysis cannot be made [44]. Neoadjuvant and adjuvant strategies often have different goals, such as downstaging patients before surgery and preventing postoperative recurrence, respectively, and thus we deemed it necessary to investigate the effect of adjuvant TACE separately from any preoperative interventions [11]. On the other hand, Zhong et al. set similar inclusion criteria to ours but utilized risk ratios as effect estimates to summarize the study outcomes [42]. This practice is considered suboptimal when dealing with time-to-event outcomes, such as OS and RFS, since censored patients lost during follow-up may result in heavily biased and inaccurate 2 × 2 tables [56]. The current study essentially represents an update on the previous meta-analysis by Zhong et al. [42], ensuring that the results of newer RCTs published since then have been included and that appropriate effect measures have been utilized to summarize long-term survival outcomes.

The rationale behind using TACE in the adjuvant setting is to prevent the recurrence of the primary tumor via proliferation of intrahepatic micro-metastases that cannot be adequately visualized preoperatively or during resection [57]. Major abdominal surgery, such as hepatectomy, results in the release of growth factors and proinflammatory cytokines (such as macrophage inflammatory protein-2, interleukin-6, and tumor necrosis factor alpha) that promote regeneration of the remaining liver tissue but may also inadvertently enhance the proliferation of these remaining tumor cells [58,59,60]. The rapid proliferation of these cells makes them suitable targets for chemotherapeutic drugs, and locoregional therapies such as TACE ensure that that these agents achieve adequate concentrations in the target tissue, while also minimizing their systemic toxicity [37]. Nonetheless, TACE remains an invasive procedure, associated with adverse effects and complications of varying severity, as well as increased cost of care [61]. Appropriately selecting patients that will most likely benefit from TACE is therefore imperative in order to optimize outcomes without exposing patients to unnecessary therapeutic interventions. A variety of prognostic tools have been developed over the years for this purpose, utilizing clinical variables (e.g., tumor number and size) [62,63], laboratory values (e.g., AFP, γ-glutamyl transferase levels) [64,65], molecular biomarkers (e.g., Ki67, microRNA-1268a, cochlin) [66,67,68], immunologic parameters (e.g., CD8+ cell infiltration and programmed death-ligand 1 expression) [69], and circulating tumor cells [70]. However, the external validation and comparison of these tools in a prospective setting remains an unmet need. Ideally, this would lead to the establishment of uniform criteria for the selection of patients undergoing hepatectomy who would most likely benefit from adjuvant TACE, thus optimizing patient outcomes and resource utilization. Although we attempted to investigate the potential impact of multiple clinical variables on the outcomes of TACE through subgroup and meta-regression analyses, the scarcity of data precluded any meaningful conclusions and highlighted the need for a uniform checklist of baseline patient characteristics that should be reported in future RCTs on resectable HCC.

The majority of HCC worldwide is attributable to viral hepatitis, mainly chronic HBV and HCV infection. This relationship is reflected in the prevalence of HCC among areas with different viral hepatitis burdens: countries in Eastern Asia, where HBV is endemic and congenital transmission is common, tend to have much higher HCC rates compared to Western countries, where HBV infection is sporadic [71,72]. Consistently with this trend, the vast majority of patients included in our analysis were HBV-positive (902/1121, 80.5% based on available data), as our study sample was geographically restricted to Eastern Asia, with 90% (9/10) of all included studies originating from China. In comparison, the HBV infection rate in HCC patients in the United States is just 10%–15% [73]. Differences in the epidemiology and transmission of viral hepatitis among various geographical regions also translate to differences in the demographic characteristics of HCC patients. In particular, there is a stronger predominance for males and younger ages in areas where viral hepatitis and HCC are highly prevalent [73]. This pattern was also observed in our sample, as males comprised the majority of patients (976/1122, 87.0%), whereas the mean age was just 51.0 ± 11.0 years. As a result of the significant differences in terms of HBV prevalence, age, and gender among patients in Eastern and Western countries, the applicability of these results may be limited to countries with similar epidemiological and transmission dynamics to countries such as China. Future clinical trials should target diverse populations through a multinational, multicenter design to address the epidemiological differences between different HCC patient subgroups.

The results of this study should be interpreted with caution due to some of its inherent limitations. Despite limiting our inclusion criteria to RCTs with the goal of summarizing the highest level of evidence on the topic, our risk of bias assessment revealed that there were significant concerns for biased results in all of the included trials. There was significant variation in the reporting practices of epidemiologic data between authors, and there was limited data availability for almost all variables included in our exploratory analyses on potential confounders and effect modifiers. Similarly, there was variable reporting and considerable heterogeneity in the TACE technique, agents, and dosing schedule. Therefore, none of these analyses had adequate statistical power to detect the potential influence of these variables on the outcomes. The recommended number of studies for each variable included in a meta-regression model is 10, yet none of our analyses was able to fully meet this expectation [74]. Nonetheless, we found some very weak trends, albeit non-statistically significant, between both OS and RFS and the proportion of anatomic to parenchymal-sparing resection. These associations warrant careful interpretation and may prompt further investigation in future clinical trials and meta-analyses once more data become available. There was also significant heterogeneity among the included trials regarding the TACE regimen and frequency used. The chemotherapy component of TACE in particular varied widely and was either a single agent or a combination of different chemotherapeutic agents with varying mechanisms of action. The frequency of TACE also varied widely both within and between the included trials and ranged from a single TACE course up to five courses, which were spaced as close as 2 weeks or as far as 8 weeks apart. The current evidence suggests that the choice of chemotherapeutic agent for TACE may not significantly influence patient outcomes, as a recent meta-analysis showed that anthracyclines and platinum-based agents led to similar outcomes [75]. Meanwhile, selecting the number of TACE courses is a more complex issue of balance between minimizing tumor recurrence and preserving liver function [76]. Other aspects related to the standardization and optimization of the TACE protocol, such as choosing the embolic agent or combining TACE therapies with other adjuvant agents are yet to be explored in an RCT setting. Novel embolic agents, such as DC microsphere beads, may enable more precise dosing and the extended release of the chemotherapeutic agents loaded onto them [77], whereas the proangiogenic response following TACE may suggest a potential synergy with anti-angiogenic agents [78]. Finally, as discussed above, the applicability of our results is limited, as almost all studies in our sample originated from China. Therefore, our findings may not be generalizable outside of Eastern Asia, where HBV transmission dynamics and HCC patient demographics may significantly differ.

## 5. Conclusions

In conclusion, adjuvant TACE following hepatectomy may be associated with an oncologic benefit in Eastern Asian patients with resectable HCC. Even though multiple RCTs have been conducted to date on this topic, the overall level of evidence remains low, as these studies have a high risk of bias. In addition, these trials are all geographically limited to Eastern Asia and their results may not be directly applicable to patients outside this geographical setting, because of substantial epidemiological differences. The significant variability in the TACE protocols used further complicates the clinical decision-making process and highlights the need for standardization of the technique. High-quality multi-national RCTs would be needed before adjuvant TACE can be implemented in daily clinical practice as the new standard of care for patients undergoing hepatectomy.

## Figures and Tables

**Figure 1 cancers-13-02984-f001:**
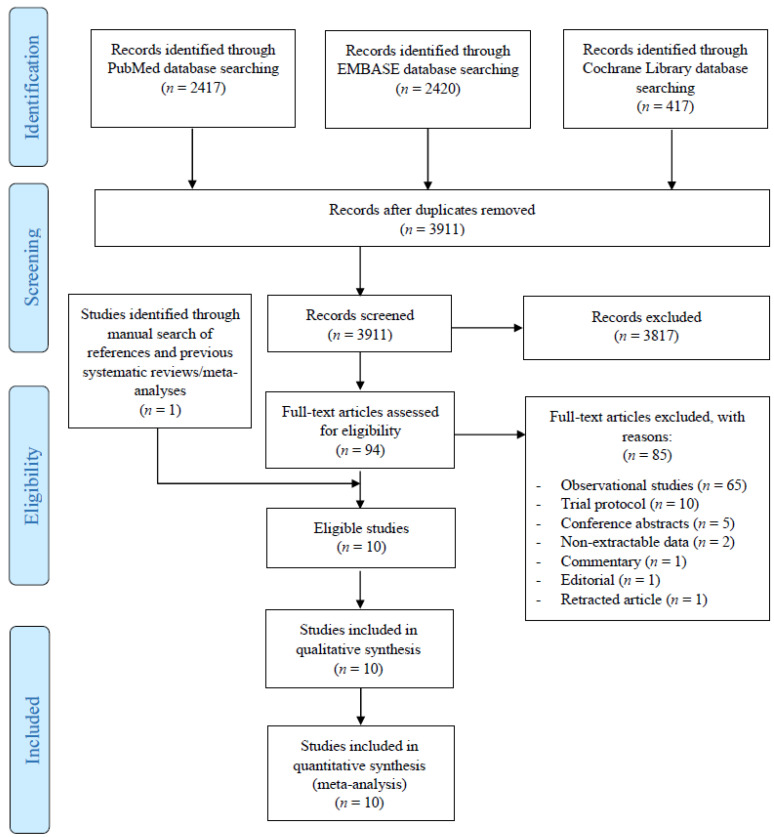
Preferred Reporting Items for Systematic Reviews and Meta-Analyses (PRISMA) flow diagram of the study selection process.

**Figure 2 cancers-13-02984-f002:**
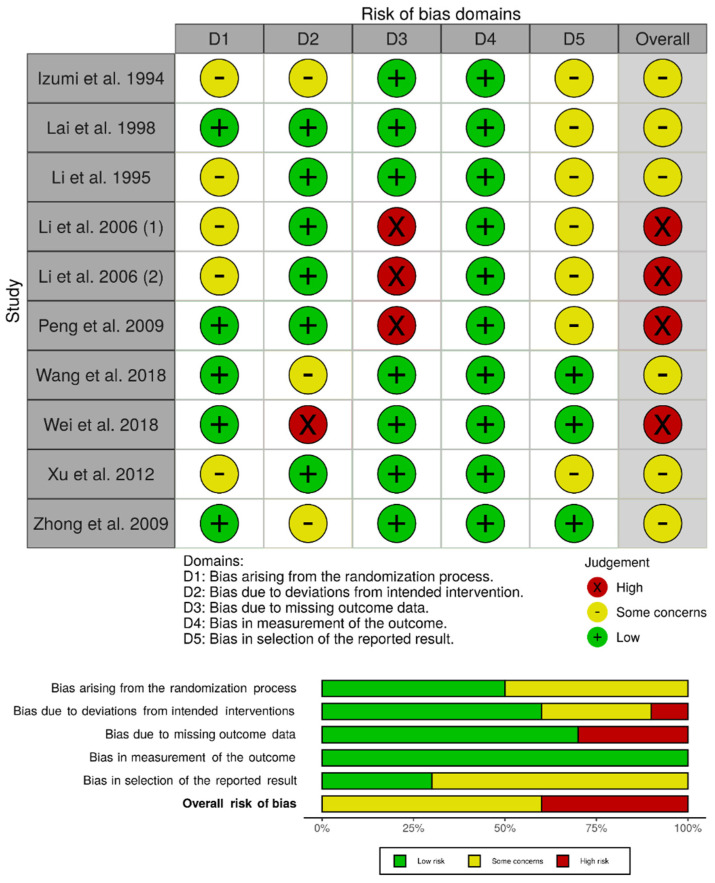
Risk of bias assessment for observational studies using the Risk of Bias 2.0 tool.

**Figure 3 cancers-13-02984-f003:**
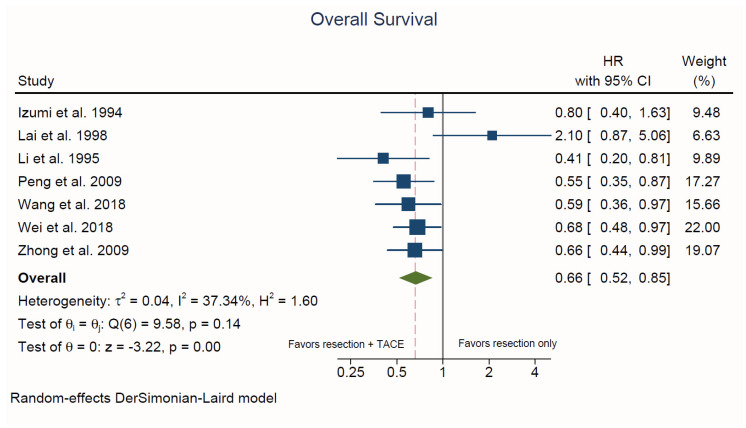
Forest plot of overall survival. HR: hazard ratio; CI: confidence interval.

**Figure 4 cancers-13-02984-f004:**
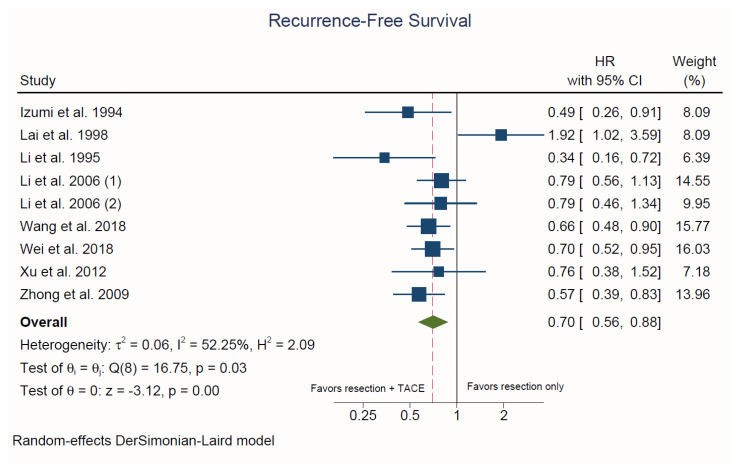
Forest plot of recurrence-free survival. HR: hazard ratio; CI: confidence interval.

**Table 1 cancers-13-02984-t001:** Characteristics of the included studies.

Study	Country	Adjuvant TACE vs. Hepatectomy Only	Males	Age (Years)	CPS (A/B/C)	HBV (+)	Tumor Size (cm)	MVI	Anatomic Resection	TACE Regimen	TACE Frequency
Izumi 1994 [30]	Japan	23 vs. 27	44/50	63.6 ± 9.9	N/A	8/50	N/A	50/50	21/50	Lipidol, doxorubicin, mitomycin	Once
Lai 1998 [32]	China	30 vs. 36	53/66	54.0 ± 12.0	N/A	56/66	9.5 ± 11.6	N/A	60/66	Lipiodol, cisplatin	3 courses every 2 months
Li 1995 [31]	China	47 vs. 47	N/A	N/A	N/A	N/A	N/A	N/A	N/A	Lipiodol, doxorubicin, mitomycin	1–3 courses every 4–6 weeks
Li 2006 (1) [33]	China	35 vs. 37	65/72	54.4 ± 10.6	33/39/0	63/71	9.8 ± 3.2	72/72	72/72	Lipiodol, doxorubicin, mitomycin, cisplatin, carboplatin	3 courses every 2 weeks
Li 2006 (2) [34]	China	39 vs. 45	73/84	51.6 ± 10.6	45/39/0	69/84	5.2 ± 1.7	N/A	84/84	Lipiodol, doxorubicin, mitomycin, cisplatin, carboplatin	3 courses every 2 weeks
Peng 2009 [35]	China	51 vs. 53	96/104	48.2 ± 11.2	90/14/0	71/104	8.7 ± 2.7	104/104	35/104	Lipiodol, 5-fluorouracil, doxorubicin	2–5 courses every 1–2 months
Wang 2018 [36]	China	140 vs. 140	230/280	53.4 ± 10.0	N/A	280/280	N/A	165/280	121/280	Lipidol, doxorubicin	N/A
Wei 2018 [37]	China	116 vs. 118	212/234	46.3 ± 9.4	232/2/0	195/234	N/A	234/234	N/A	Lipiodol, epirubicin, mitomycin, carboplatin	N/A
Xu 2012 [38]	China	59 vs. 58	101/117	50.7 ± 9.5	N/A	55/117	3 ± 1.2	9/117	N/A	Lipiodol, epirubicin, 5-fluoruracile hydroxycamptothecin	N/A
Zhong 2009 [39]	China	57 vs. 58	102/115	47.9 ± 10.8	114/1/0	105/115	9.6 ± 3.7	48/115	86/115	Lipiodol, carboplatin, mitomycin, epirubicin	N/A

CPS: Child–Pugh score; N/A: not available; MVI: microvascular invasion; TACE: transarterial chemoembolization.

## Data Availability

No new data were created or analyzed in this study. Data sharing is not applicable to this article.

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
