# Peer review of "Adjuvant Transarterial Chemoembolization Following Curative-Intent Hepatectomy Versus Hepatectomy Alone for Hepatocellular Carcinoma: A Systematic Review and Meta-Analysis of Randomized Controlled Trials"

_cancers, 2021, doi:10.3390/cancers13122984_

Round 1
Reviewer 1 Report
This systematic review and meta-analysis from Esagian and colleagues focuses on the impact of adjuvant TACE after surgical resection for HCC.
The methodology is rigorous and the results are intriguing
Study limitations (i.e. limitation of these results to eastern population) are well discussed as well.
I would like the Authors to consider citing this paper in the introduction section: https://doi.org/10.3390/cancers13071671
Best regards
Author Response
Thank you very much for your feedback. Following your suggestion, we now cite this paper in the Introduction section (line 55).
Reviewer 2 Report
The aim of this review “was to clarify the role of adjuvant transarterial chemoembolization (TACE) for patients with resectable hepatocellular carcinoma (HCC) undergoing hepatectomy”. The authors performed a systematic review of the available high-quality randomized controlled trials (RCTs) and synthesized their data using the methodology of meta-analysis. Ten eligible RCTs reporting on 1216 patients were identified. The combination of hepatectomy and adjuvant TACE was associated with superior overall survival (OS) (hazard ratio [HR]: 0.66, 95% confidence interval [CI]: 0.52 to 0.85; p < 0.001) and recurrence-free survival (RFS) (HR: 0.70, 95% 29 CI: 0.56 to 0.88; p < 0.001) compared to hepatectomy alone.The presented meta-analysis concerns an important problem, not fully solved, especially in view of the variety of patients studied, epidemiological data, and the prognostic role of the applied therapy.
The aim of the paper is clear, although there are already many similar meta-analyses, i.e. on the role of adjuvant TACE for resectable HCC (ref. no. 39-50). However only two meta-analyses limited their inclusion criteria to RCTs (ref. 40, 42).
This paper feels like an update of the previous meta-analysis conducted by Zhong et al [42], ensuring that the results of more recent RCTs published since then are included and that appropriate effect measures are used to summarise long-term survival outcomes. Although the authors addressed a not entirely novel medical problem, they confirmed or updated previous conclusions based on more recent research findings. The authors found some very weak trends, albeit non-statistically significant, between both OS and RFS and the proportion of anatomic to parenchymal-sparing resection. They also provided directions for further research.
The aim of the work has been achieved and the work limits are marked.
The Material and Methods are described clearly and this part of the paper is not objectionable.
The results of the work are also presented correctly, legibly and do not raise any objections.
Small errors and/or comments: 1. Line 170 – statistical significance was defined as p< 0.1, should be „p <0.01”.
- It seems to me that some of the data in Figures 3 and 10 (supplement data) and Figures 4 and 11 (supplement data) are unnecessarily doubled. It is better to put 10 and 11 in the main text; or please explain possibly why this is so.
3. Please clarify what "control group" means in subsection 3.4.3. (line 242) and Table 1 (TACE vs. control), is control group means the hepatectomy itself, so no TACE therapy? Is "control group" the correct term? Please check the size of the study groups, e.g. in the paper by Li J, 1995 [ref 28] 140 patients were studied and in Table 1 it is 47 + 47, why the discrepancies? Please check the size of the study groups throughout paragraph 3.4.3. as it is not clear. 4. References need to be corrected in accordance with the journal's recommendation, especially with regard to textbooks cited, and in journal articles - to standardise the volumes, pages , no. doi given. I do not understand many citations, e.g. no. 17 (not enough data), 28 (which names relate to the authors?), 66 (not enough details).
Author Response
We appreciate the reviewer for their feedback.
Comment 2.1: “Line 170 – statistical significance was defined as p< 0.1, should be „p <0.01”.”
Response: Thank you for your comment. The statistical significance the reviewer is referring to refers to the test for statistical heterogeneity. In the seminal paper describing their methodology, Egger and colleagues used a p < 0.1 value for statistical significance, which has been traditionally used when assessing statistical heterogeneity in a meta-analysis (1).
Comment 2.2: “It seems to me that some of the data in Figures 3 and 10 (supplement data) and Figures 4 and 11 (supplement data) are unnecessarily doubled. It is better to put 10 and 11 in the main text; or please explain possibly why this is so.”
Response: Thank you for your comment. Figures 10 and 11 represent subgroup analyses of the OS and RFS, respectively, according to the risk of bias of each individual study included in this meta-analysis. These analyses serve as sensitivity analyses that assess the potential influence of factors such as the risk of bias on the validity of our results. Hence, Figures 10 and 11 represent exploratory secondary analyses (rather than duplicate analyses) and were placed in the Supplementary Material, while Figures 3 and 4 represent our primary analyses and were included in the Main Text.
Comment 2.3: “Please clarify what "control group" means in subsection 3.4.3. (line 242) and Table 1 (TACE vs. control), is control group means the hepatectomy itself, so no TACE therapy? Is "control group" the correct term?”
Response: Thank you for your comment. We have replaced the term “control group” with “hepatectomy only” to improve the clarity of our manuscript.
Comment 2.4: “Please check the size of the study groups, e.g. in the paper by Li J, 1995 [ref 28] 140 patients were studied and in Table 1 it is 47 + 47, why the discrepancies”
Response: Thank you for your comment. Li et al. compared the efficacy of adjuvant TACE vs. hepatectomy in two separate patient cohorts, including 94 (47+47) patients receiving hepatectomy with curative intent and 46 patients (23+23) receiving palliative hepatectomy. Due to the substantial differences in prognosis that could confound our findings, we limited our inclusion criteria to patients receiving hepatectomy with curative intent only, as defined in the 2.1. Study Design and Inclusion/Exclusion Criteria section (line 88). Therefore, we only extracted data for the patient cohort receiving hepatectomy with curative intent (94 patients), rather than the total patient population (140 patients).
Comment 2.5: “Please check the size of the study groups throughout paragraph 3.4.3. as it is not clear.”
Response: Thank you for your comment. As noted in the beginning of the paragraph, the sizes of each group are calculated by adding the populations of those studies that provided data on this outcome. Only 4 out of the 10 included studies provided this information, and therefore the sample sizes of each group are only a fraction of the total sample size of our meta-analysis. We have added the following phrase in line 292: “Pooling the 4 studies,” to improve the clarity of our text and avoid any confusion regarding the sample sizes for this outcome.
Comment 2.6: “References need to be corrected in accordance with the journal's recommendation, especially with regard to textbooks cited, and in journal articles - to standardise the volumes, pages , no. doi given. I do not understand many citations, e.g. no. 17 (not enough data), 28 (which names relate to the authors?), 66 (not enough details).”
Response: Thank you for your comment. We have updated the formatting of our citations to match the Journal’s proposed style. Citation #17 (#20 in the revised manuscript) is a software program, while citation #66 (#72 in the revised manuscript) is a website and therefore do not follow the same format as journal article citations. We have also corrected the information of citation #28 (#31 in the revised manuscript).
References
- Egger M, Smith GD, Schneider M, Minder C. Papers Bias in meta-analysis detected by a simple, graphical test.
Reviewer 3 Report
Authors have made a valuable work by performing this metaanalysis. Treatment of most liver tumors is multi-modular, however, treatment of HCC is clasically the most complicated one, involving a large number of possible surgical and interventional options. TACE was previously a technique used for non-resectable HCC's, but this metaanalysis provides good quality evidence, that it can be used prior to liver resection to increase the overall survival and reccurence free survival rates after treatment. These findings that are obtained from multiple studies might influence future strategies for HCC treatment, therefore this manuscript should be considered for acceptance.
Author Response
Thank you very much for your kind feedback
Reviewer 4 Report
The present study by Esagian et al. is a systematic review and meta-analysis of randomized controlled trials evaluating the impact of adjuvant TACE after surgical resection of HCC. The authors found improved overall and disease-free survival in patients with combined TACE+surgery as compared with surgery alone. The manuscript is well written and informative. The methodology is adequate. The main concerns are the lack of novelty (there are previous meta-analyses with similar conclusions, some of the published just 2 years ago) and the partial evaluation of outcomes (information about safety is completely missing).
The authors are kindly invited to consider the following comments:
- As the authors have pointed out, there many several previous metaanalyses with similar aims and conclusions as the present one. Apart from those reported by the authors (refs 40 and 42), they missed more recent ones such as PMID: 28276833 (2017), PMID: 31256949 (2019), PMID: 31081401 (2019). In its present form, the manuscript does not provide a rationale for a need of an updated metaanalysis in 2021. The authors should emphasize in the discussion what is the new evidence provided in their analysis and what could be changed in clinical practice with the publication of this paper.
- Adverse events have not been analyzed in the present metaanalysis. If there is a significant heterogeneity in reporting safety outcomes across randomized trials, which I find most likely, the authors should present the excess of adverse events attributable to TACE in the randomized trials included as a qualitative synthesis. In its present form the meta-analysis offers a partial viewpoint and safety information would be critical to make adequate clinical decisions.
- Another important limitation is that the evaluated intervention (TACE) is heterogenoeus across different RCTs. Indeed, TACE protocols varied widely regarding the number of sessions, timing and even the drug infused. This makes the results of the metaanalysis hardly interpretable and this should be highlighted by the authors as a major limitation.
- It is surprising that all included studies are based on eastern population (9 out of 10 were performed in China). The authors have adequately highlighted this as a limitation. In opinion of the authors, What is the main reason for the lack of randomized trials in the western world?
- In the table 1, there is critical information missing from some studies including the diameter of the larger nodule, the presence of microvascular invasion… Is this information not provided in the original manuscripts? Did the authors make an effort to contact the authors of the RCTs to retrieve this critical information?
Author Response
We appreciate the reviewer for providing feedback on our work. We hope after following your recommendations, our revised manuscript is now more clear and comprehensive.
Comment 4.1: “As the authors have pointed out, there many several previous metaanalyses with similar aims and conclusions as the present one. Apart from those reported by the authors (refs 40 and 42), they missed more recent ones such as PMID: 28276833 (2017), PMID: 31256949 (2019), PMID: 31081401 (2019). In its present form, the manuscript does not provide a rationale for a need of an updated metaanalysis in 2021. The authors should emphasize in the discussion what is the new evidence provided in their analysis and what could be changed in clinical practice with the publication of this paper.”
Response: Thank you for your comment. We present comprehensive list of all previous meta-analyses on the topic (lines 276-277) in the Discussion section. All 3 articles mentioned have already been included in our original citation list: PMID: 28276833 is citation #47, PMID: 31256949 is citation #44, PMID: 31081401 is citation #51 in the original manuscript.
The second paragraph of our Discussion provides a detailed explanation as to why there is a need for an updated meta-analysis on the topic. The majority of previous meta-analyses have included observational studies in the inclusion criteria, which may severely compromise the validity of their results due to the inherent risk of bias associated with them. The only previous meta-analyses to limit their inclusion criteria to randomized controlled trials, which is considered to be the best practice when conducting a meta-analysis (1,2), are Cheng et al. (3) and Zhong et al. (4). The former included studies utilizing both adjuvant and neoadjuvant TACE, leading to unclear conclusions as these modalities are used in different patient settings i.e., preventing post-surgical recurrence and preoperative tumor downstaging, respectively. Thus, the only previous meta-analysis that is truly similar and comparable to ours is the study by Zhong et al. (4), published 11 years ago. Since then, large scale RCTs have been conducted (e.g., Wei 2018 (5), Wang 2018 (6)) and as the role of TACE following hepatectomy with curative intent remains unclear to this day, we felt that conducting an updated meta-analysis was deemed appropriate.
Comment 4.2: “Adverse events have not been analyzed in the present metaanalysis. If there is a significant heterogeneity in reporting safety outcomes across randomized trials, which I find most likely, the authors should present the excess of adverse events attributable to TACE in the randomized trials included as a qualitative synthesis. In its present form the meta-analysis offers a partial viewpoint and safety information would be critical to make adequate clinical decisions.”
Response: Thank you for your comment. After revisiting all articles and exhaustively extracting all data pertaining to adverse events attributable to TACE, we have added a paragraph in our Results section (3.3.4. Adverse events), as well as a phrase in Methods section (lines 136-138) to explain the definitions we used. Unfortunately, the heterogeneity of definitions and lack of standardization in the reporting methods by the authors precluded any meaningful qualitative or quantitative data synthesis for most adverse events.
Comment 4.3: “Another important limitation is that the evaluated intervention (TACE) is heterogenoeus across different RCTs. Indeed, TACE protocols varied widely regarding the number of sessions, timing and even the drug infused. This makes the results of the metaanalysis hardly interpretable and this should be highlighted by the authors as a major limitation.”
Response: Thank you for your comment. Even though the significant variation of the TACE protocols heterogeneity is presented in detail in the limitations section of our manuscript (lines 355-364 in the original manuscript), we have added the following phrase in our Conclusion to emphasize its impact in the clinical decision-making process: “The significant variability in the TACE protocols used further complicates the clinical decision-making process and highlights the need for standardization of the technique.” (lines 401-403).
Comment 4.4: “It is surprising that all included studies are based on eastern population (9 out of 10 were performed in China). The authors have adequately highlighted this as a limitation. In opinion of the authors, What is the main reason for the lack of randomized trials in the western world?.”
Response: We thank the reviewer for this comment. We believe that the increased incidence of HCC in Southeast Asia is likely the main reason that most trials were conducted in China. There could also be some additional logistics obstacles related to running RCTs in the Western countries.
Comment 4.5: “In the table 1, there is critical information missing from some studies including the diameter of the larger nodule, the presence of microvascular invasion… Is this information not provided in the original manuscripts? Did the authors make an effort to contact the authors of the RCTs to retrieve this critical information?”
Response: Thank you for your comment. Unfortunately, the data availability on these crucial parameters was limited and we were unable to retrieve any further information beyond what is presented. As many of the studies included in this analysis were conducted prior to 2000, we were unable to locate any e-mail addresses for multiple authors. Thus, we did not make any attempts to contact authors for additional data, as we felt that that selectively doing so would impart some degree of bias to our results. We have also highlighted this as one of the main limitations of our study (lines 343-345 in the original manuscript).
References
- Shea BJ, Reeves BC, Wells G, Thuku M, Hamel C, Moran J, et al. AMSTAR 2: A critical appraisal tool for systematic reviews that include randomised or non-randomised studies of healthcare interventions, or both. BMJ. 2017 Sep 21;358:4008.
- McKenzie J, Brennan S, Ryan R, Thomson H, Johnston R, Thomas J. Chapter 3: Defining the criteria for including studies and how they will be grouped for the synthesis. In: Higgins J, Thomas J, Chandler J, Cumpston M, Li T, Page M, et al., editors. Cochrane Handbook for Systematic Reviews of Interventions version 62 (updated February 2021). Cochrane; 2021.
- Cheng X, Sun P, Hu QG, Song ZF, Xiong J, Zheng QC. Transarterial (chemo)embolization for curative resection of hepatocellular carcinoma: A systematic review and meta-analyses. J Cancer Res Clin Oncol. 2014;140(7):1159–70.
- Zhong JH, Li LQ. Postoperative adjuvant transarterial chemoembolization for participants with hepatocellular carcinoma: A meta-analysis. Hepatol Res. 2010;40(10):943–53.
- Wei W, Jian PE, Li SH, Guo ZX, Zhang YF, Ling YH, et al. Adjuvant transcatheter arterial chemoembolization after curative resection for hepatocellular carcinoma patients with solitary tumor and microvascular invasion: A randomized clinical trial of efficacy and safety. Cancer Commun. 2018;38(1):1–12.
- Wang Z, Ren Z, Chen Y, Hu J, Yang G, Yu L, et al. Adjuvant transarterial chemoembolization fo hbv-related hepatocellular carcinoma after resection: A randomized controlled study. Clin Cancer Res. 2018;24(9):2074–81.
Reviewer 5 Report
This is a well organized review and meta-analysis of randomized controlled trial focusing on adjuvant transarterial chemoembolization following curative-intent hepatectomy versus hepatectomy alone for hepatocellular carcinoma. Figures are nice and clear.
I would suggest few changes to to make the study more appeal for publication.
Criticisms:
- At line 67 in the introduction section I read radioembolization with I 131 [14]. To this regard I would suggest to cite also the possible radioembolization with Y90, inserting a specific references. Please check and comment this point.
- In the Discussion section the sentences regarding the neoadjuvant TACE should be deleted, they are poor appropriate in this setting.
- In the Discussion section, at line 296 the Authors reported that: Major abdominal surgery, such as hepatectomy, results in the release of growth factors that promote regeneration of the remaining liver tissue but may also inadvertently enhance the proliferation of these remaining tumor cells. This concept should be better sustained indicating the biological pathway/s and the growth factors implicated in the growth of cancer cells. To this regard specific reference/s should be included at the end of the above sentence and added in the References section.
- In the Discussion section, and in this context no discussion on the potential pro-angiogenic rebound of TACE has been developed. It is well demonstrated that TACE induces hypoxia that in turn induces HIF-1 and HIF-2 expression and finally with VEGF production. I suggest to added this concept and to substain it with the following reference: Vascular endothelial growth factor and tryptase changes after chemoembolization in hepatocarcinoma patients Ranieri G et al. World J Gastroenterol. 2015 May 21;21(19):6018-25.
- In the Discussion section the role of more recent TACE treatment namely DEB-DOXO as possible adjuvant TACE should be briefly discussed (just a sentence), please see and cite the following Review: Trans-arterial chemoembolization as a therapy for liver tumours: New clinical developments and suggestions for combination with angiogenesis inhibitors. Gadaleta CD et al. Crit Rev Oncol Hematol. 2011 Oct;80(1):40-53.
Author Response
We would like to thank the reviewer for their feedback.
Comment 5.1: “At line 67 in the introduction section I read radioembolization with I 131 [14]. To this regard I would suggest to cite also the possible radioembolization with Y90, inserting a specific references. Please check and comment this point.”
Response: Thank you for your comment. We have added the following phrase in the Introduction section following your recommendation: “Radioembolization with Y90 is increasingly used for unresectable or borderline resectable tumors as well” (lines 69-70).
Comment 5.2: “In the Discussion section the sentences regarding the neoadjuvant TACE should be deleted, they are poor appropriate in this setting.”
Response: Thank you for your comment. Neoadjuvant TACE is only mentioned in the Discussion so as to make a distinction of our meta-analysis with previous meta-analyses on the same topic and provide a rationale for our inclusion criteria being limited to TACE in the adjuvant setting only, contrary to the meta-analysis by Cheng et al. that included studies on both adjuvant and neoadjuvant TACE (4). As pointed out by Reviewer’s #2, we feel that making this distinction is crucial to establish the rationale for an updated meta-analysis, despite numerous previous meta-analyses that have been published on the topic over the years.
Comment 5.3: “In the Discussion section, at line 296 the Authors reported that: Major abdominal surgery, such as hepatectomy, results in the release of growth factors that promote regeneration of the remaining liver tissue but may also inadvertently enhance the proliferation of these remaining tumor cells. This concept should be better sustained indicating the biological pathway/s and the growth factors implicated in the growth of cancer cells. To this regard specific reference/s should be included at the end of the above sentence and added in the References section.”
Response: We would like to thank the reviewer for pointing this out. Following your recommendation, we have added the following phrase to our manuscript: “factors and proinflammatory cytokines (such as macrophage inflammatory protein-2, interleukin-6, and tumor necrosis factor alpha)”(lines 315-316) and listed 3 relevant references to support it.
Comment 5.4: “In the Discussion section, and in this context no discussion on the potential pro-angiogenic rebound of TACE has been developed. It is well demonstrated that TACE induces hypoxia that in turn induces HIF-1 and HIF-2 expression and finally with VEGF production. I suggest to added this concept and to substain it with the following reference: Vascular endothelial growth factor and tryptase changes after chemoembolization in hepatocarcinoma patients Ranieri G et al. World J Gastroenterol. 2015 May 21;21(19):6018-25.”
Comment 5.5: “In the Discussion section the role of more recent TACE treatment namely DEB-DOXO as possible adjuvant TACE should be briefly discussed (just a sentence), please see and cite the following Review: Trans-arterial chemoembolization as a therapy for liver tumours: New clinical developments and suggestions for combination with angiogenesis inhibitors. Gadaleta CD et al. Crit Rev Oncol Hematol. 2011 Oct;80(1):40-53..”
Response: Thank you for your comments. We added the following sentences to our Discussion section, citing both articles you mentioned: “Other aspects related to the standardization and optimization of the TACE protocol, such as choosing the embolic agent or combining therapies TACE with other adjuvant agents are yet to be explored in an RCT setting. Novel embolic agents as the DC microsphere beads may enable more precise dosing and extended release of the chemotherapeutic agents loaded onto them, while the proangiogenic response following TACE may suggest a potential synergy with anti-angiogenic agents.” (lines 384-390).
Round 2
Reviewer 4 Report
The authors have made an effort to answer my previous concerns. Most of the queries have been adequately answered or acknowledged as limitations.